# Whole-exome sequencing in a Japanese multiplex family identifies new susceptibility genes for intracranial aneurysms

**Tatsuya Maegawa[1,2], Hiroyuki Akagawa[1]\*, Hideaki Onda[2,3], Hidetoshi Kasuya[2]**

**1** Tokyo Women's Medical University Institute for Integrated Medical Sciences (TIIMS), Tokyo, Japan,
**2** Department of Neurosurgery, Tokyo Women's Medical University Medical Center East, Tokyo, Japan,
**3** Division of Neurosurgery, Kofu Neurosurgical Hospital, Kofu, Yamanashi, Japan

\* akagawa.hiroyuki@twmu.ac.jp

## Abstract

### Background

Intracranial aneurysms (IAs) cause subarachnoid hemorrhage, which has high rates of mortality and morbidity when ruptured. Recently, the role of rare variants in the genetic background of complex diseases has been increasingly recognized. The aim of this study was to identify rare variants for susceptibility to IA.

### Methods

Whole-exome sequencing was performed on seven members of a Japanese pedigree with highly aggregated IA. Candidate genes harboring co-segregating rare variants with IA were re-sequenced and tested for association with IA using additional 500 probands and 323 non-IA controls. Functional analysis of rare variants detected in the pedigree was also conducted.

### Results

We identified two gene variants shared among all four affected participants in the pedigree. One was the splicing donor c.1515+1G>A variant in *NPNT* (Nephronectin), which was confirmed to cause aberrant splicing by a minigene assay. The other was the missense p.P83T variant in *CBY2* (Chibby family member 2). Overexpression of p.P83T CBY2 fused with red fluorescent protein tended to aggregate in the cytoplasm. Although Nephronectin has been previously reported to be involved in endothelial angiogenic functions, CBY2 is a novel molecule in terms of vascular pathophysiology. We confirmed that *CBY2* was expressed in cerebrovascular smooth muscle cells in an isoform2-specific manner. Targeted *CBY2* re-sequencing in additional case-control samples identified three deleterious rare variants (p.R46H, p.P83T, and p.L183R) in seven probands, showing a significant enrichment in the overall probands (8/501) compared to the controls (0/323) ($p$ = 0.026, Fisher's extract test).

**Data Availability Statement:** All relevant data are within the manuscript and its Supporting information files. The minimal data set of the

whole-exome sequencing was provided in the Supporting Information S1 Table.

**Funding:** This work was supported by JSPS KAKENHI (grant no. 19K18444 (T.M.), 15K10316, and 18K08953 (H.K.)). The funders had no role in study design, data collection and analysis, decision to publish, or preparation of the manuscript.

**Competing interests:** The authors have declared that no competing interests exist.

## Conclusions

*NPNT* and *CBY2* were identified as novel susceptibility genes for IA. The highly heterogeneous and polygenic architecture of IA susceptibility can be uncovered by accumulating extensive analyses that focus on each pedigree with a high incidence of IA.

## Introduction

An intracranial aneurysm (IA) is an abnormal ballooning that usually develops at the bifurcation of the major cerebral arteries. Its rupture can cause subarachnoid hemorrhage (SAH), one of the most severe forms of stroke, which has high rates of mortality and morbidity. IA is a common complex trait with a prevalence of 3.2% in the general population [1]. In addition to well-established risk factors, such as smoking and hypertension [2], genetic predisposition has been demonstrated to play important roles in the pathophysiology of IA through a number of genome-wide association studies (GWASs) [3, 4]. In recent international GWASs on strokes, tens to hundreds of thousands of patients and controls were recruited, among which the study group for IA explained over half of the disease heritability of IA, including 17 risk loci [5, 6]. The remaining proportion of heritability, the so-called missing heritability, can be partly explained by the effects of rare variants that cannot be detected through standard genetic association analysis, such as GWASs, which targets common polymorphisms [7]. Whole-exome sequencing (WES) approaches are now widely used for detecting rare susceptibility variants for complex traits, as well as the causative mutations of Mendelian disorders by analyzing families with multiple affected members. To date, family-based WES studies for IA have proposed several genes harboring rare susceptibility variants with allelic frequencies less than 1% in the general population, such as *TSHD1* and *ANGPTL6*, which underwent functional analyses of the genes and/or their variants in relation to the cerebrovascular pathophysiology [8, 9]. However, much is still unknown about the role of rare variants in the genetic architecture of IA, especially in patients from East Asia. In the present study, we attempted to identify novel susceptibility genes attributed to functional rare variants, starting with a single Japanese family with highly aggregated IA.

## Materials and methods

### Ethics statement

The Ethics Committee of Tokyo Women's Medical University approved the study protocol. All participants or guardians of those who suffered neurological deficits after aneurysmal subarachnoid hemorrhage provided written informed consent.

### Subjects

In the past two decades, we performed linkage and association studies in a total of 90 nuclear families, including siblings affected by IA [10–12]. During the long-term follow-up of these families, aneurysmal SAH arose in the offspring of an affected sibling trio (Fig 1A). This multiplex family (F2054) had one of the largest numbers of patients with IA among the 90 families we studied. Therefore, in the present study, F2054 was extensively analyzed for novel susceptibility genes.

For re-sequencing and association analysis of candidate genes detected in the pedigree analysis of F2054, additional 500 probands with IA and 323 non-IA controls were enrolled. All

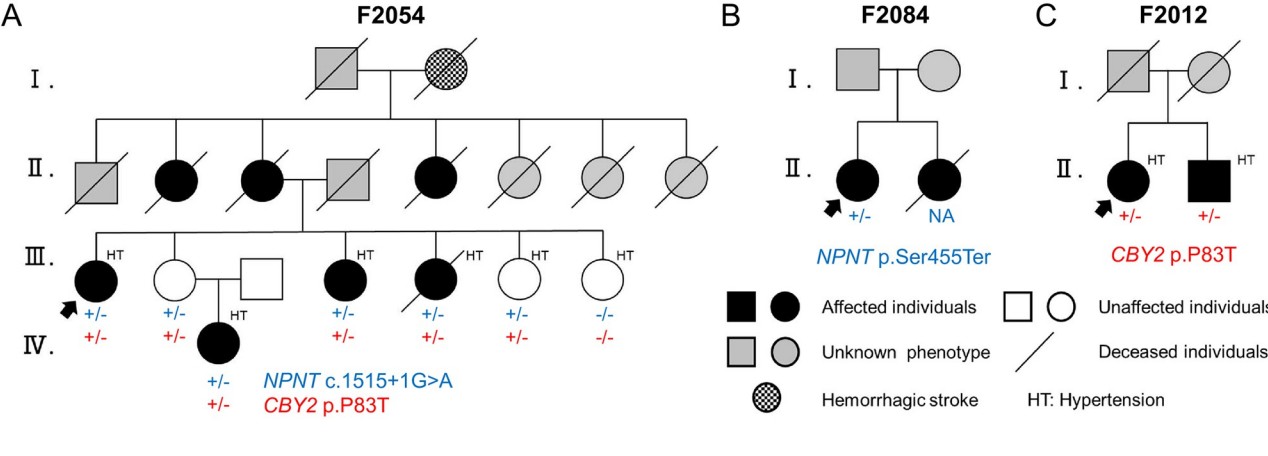

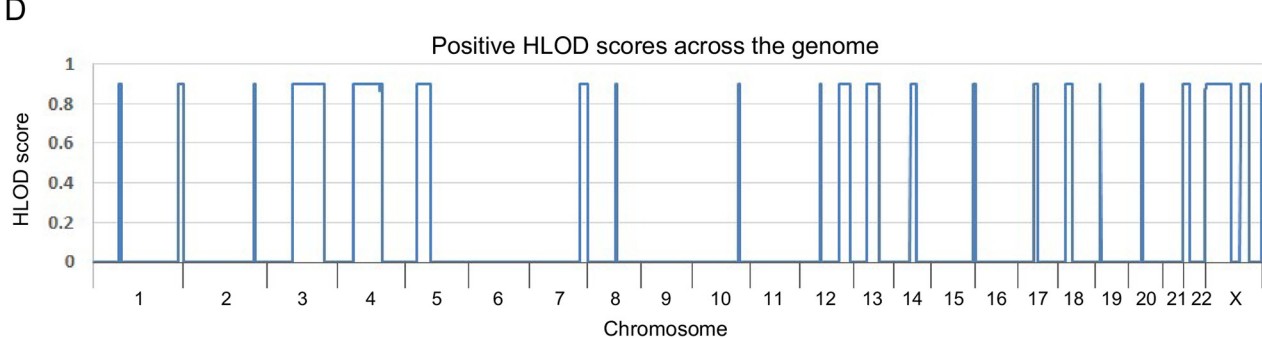

**Fig 1. Japanese multiplex families with IA.** Pedigree charts of the exome-sequenced family (A) and the Sanger-sequenced affected siblings (B, C) who carried pathogenic variants in *CBY2* and *NPNT*. Colored plus and minus signs (+ and −) indicate mutated and wild-type alleles, respectively. *CBY2* and *NPNT* were identified from the exome data using genome-wide linkage analysis in F2054 (D). Abbreviations: HLOD, maximum heterogeneity logarithm of odds.

subjects were of the Japanese ethnicity. The IA probands included 153 cases of familial IA, of which 89 were from the nuclear families we previously studied [10–12], and 347 cases of sporadic IA. The presence of IA in the patients was confirmed by conventional angiography, 3D computed tomography angiography, magnetic resonance angiography, or surgical findings. We systematically excluded dissecting IAs and IAs secondary to genetic disorders such as autosomal dominant polycystic kidney disease. The 323 unrelated controls were outpatients at the Tokyo Women's Medical University Hospital, Tokyo Women's Medical University Medical Center East, and their nearby affiliated hospitals with conditions other than IA. The controls showed no evidence of IA on computed tomography or magnetic resonance angiography, and none of them had a family history of aneurysmal SAH.

We also prepared 105 Japanese population-based controls, who were not confirmed to have no IA and were totally different from the 323 non-IA controls, in order to confirm allelic frequencies of the candidate variants in the Japanese general population.

## Genetic analysis

Genomic DNA (gDNA) was extracted from peripheral blood leukocytes using a standard method. gDNA from each of the seven participants from the pedigree F2054 (III-1 to 4, III-6, III-7, and IV-1) (Fig 1A) was subjected to exome enrichment using a SureSelect Human All Exon V5 kit, according to the manufacturer's instructions (Agilent Technologies Inc., Santa

Clara, USA). Enriched DNA libraries were sequenced using 100-bp paired-end reads on a HiSeq 2000 sequencer (Illumina, San Diego, USA). The reads were aligned to the Genome Reference Consortium Human Build 37 (GRCh37). After the first alignment step was performed using BWA-MEM in Burrows-Wheeler Aligner version 0.7.15 (https://github.com/lh3/bwa/releases/tag/v0.7.15) [13], any discordantly mapped or unmapped read pairs were realigned using Novoalign version 3.04.06 (http://www.novocraft.com). Picard version 2.5.0 (https://github.com/broadinstitute/picard/releases/tag/2.5.0) was used to mark and remove polymerase chain reaction (PCR) duplicates. Variants were identified by single-sample calling with HaplotypeCaller using Genome Analysis Toolkit version 3.5 (https://console.cloud.google.com/storage/browser/gatk-software/package-archive/gatk) [14] and annotated using ANNOVAR version 2019Dec06 (http://annovar.openbioinformatics.org/) [15].

The seven participants from the pedigree F2054 (Fig 1A) were also genotyped with a HumanOmni1-Quad BeadChip array (Illumina) for genome-wide linkage analysis, which allowed for the narrowing down of susceptibility chromosomal loci in this pedigree [16]. From the single-nucleotide polymorphism (SNP) data obtained from the array, we selected 45,525 SNPs (44,621 autosomal and 904 X-chromosomal SNPs) according to the following procedures: (i) A/T and C/G SNPs were excluded to avoid misinterpretation of the forward/reverse strand, (ii) requiring that they were heterozygous in at least three genotyped individuals, (iii) the SNPs genotyped in the HapMap JPT individuals were kept and merged together with the JPT data (https://www.sanger.ac.uk/resources/downloads/human/hapmap3.html) [17], and (iv) pairwise linkage disequilibrium (LD)-based SNP pruning was performed to remove one of a pair of SNPs if the LD ($r^2$) was greater than 0.2 using PLINK version 1.07 (http://zzz.bwh.harvard.edu/plink/index.shtml) [18]. These selected SNPs were used for a parametric linkage analysis, calculated using Merlin version 1.1.2 (http://csg.sph.umich.edu/abecasis/merlin/index.html), assuming a dominant model with reduced penetrance (set at 0.667) on the HapMap GRCh37 recombination map (ftp://ftp-trace.ncbi.nih.gov/1000genomes/ftp/technical/working/20110106_recombination_hotspots/) [17, 19]. We also analyzed allele sharing by computing identity by descent (IBD) segments using Beagle version 3.3 (https://faculty.washington.edu/browning/beagle/b3.html) from the filtered SNP array data, as mentioned above in (i) and (iii) [20].

Confirmation of the candidate variants identified by exome sequencing and subsequent re-sequencing of the candidate genes in the additional cohorts was performed via standard PCR-based amplification, followed by BigDye Terminator cycle sequencing on a 3130xl Genetic Analyzer (Thermo Fisher Scientific, Waltham, USA).

## In vitro splicing assay

One of the variants was identified in an exon-intron junction (*NPNT* c.1515+1G>A, NM_001184692), whose impact on splicing was first evaluated using various *in silico* predictors, as described in the following section. To confirm these *in silico* predictions, a minigene assay was performed [21]. The exon and its adjacent intronic sequences containing wild-type or mutant splice-site variants were subcloned into the *Xho*I/*Spe*I-digested Exontrap pET01 vector (MoBi-Tec GmbH, Göttingen, Germany). HeLa cells were cultured in Eagle's minimum essential medium containing non-essential amino acids and 10% fetal bovine serum (FBS). The cells were transfected with the wild-type or mutant vector using Lipofectamine 2000 (Thermo Fisher Scientific). At 24 h post-transfection, total RNA was extracted, and reverse transcription (RT)-PCR covering between the 5' and 3' exons in the pET01 vector was performed. The RT-PCR products were visualized by agarose gel electrophoresis and sequenced using BigDye Terminator cycle sequencing on a 3130xl Genetic Analyzer (Thermo Fisher Scientific). The primer sequences are provided in S1 File.

## Expression analysis in cerebrovascular tissues

*CBY2* (alternatively referred to as *SPERT*) was of particular interest in the genetic analysis and was therefore analyzed for its expression in cerebral arterial tissues. Immunohistochemical staining of paraffin-embedded specimens of surgically resected peripheral cerebral arteries, arteriovenous malformations (AVMs), and IA was performed using a Histofine MAX-PO kit (Nichirei Biosciences Inc., Tokyo, Japan). The antibodies used were rabbit polyclonal anti-SPERT (#AP5649a) produced using an amino-terminal region (amino acids 104–134, iso-form1) of human SPERT as an immunogen (Abgent, San Diego, USA), mouse monoclonal anti-CD34 (Nichirei Biosciences Inc.), mouse monoclonal anti-SMA (Dako, Agilent Technologies Inc.), and mouse monoclonal anti-GFAP (Nichirei Biosciences Inc.). Antigen retrieval and the dilution of antibodies were performed according to the manufacturer's instructions.

RT-PCR to amplify the *CBY2* transcripts was performed using the following complementary DNAs (cDNAs) from human tissues: Human Adult Normal Tissue cDNA from Testis and Brain (BioChain Institute, Inc., Newark, USA) as positive controls, according to the ENCODE data provided in the Expression Atlas (https://www.ebi.ac.uk/gxa/home), human brain vascular smooth muscle cell (HBVSMC) cDNA, and human brain microvascular endothelial cell (HBMEC) cDNA (ScienCell Research Laboratories Inc., Carlsbad, USA). *GAPDH* was used as an internal reference for each sample. The sequences of the primers used are provided in S1 File.

## Cell imaging

The *CBY2* transcriptional sequence encoding isoform 2 (Q8NA61-2) was obtained by purchasing a SPERT (NM_001286342) human untagged clone plasmid (OriGene Technologies Inc., Rockville, USA). The sequence variant detected in pedigree F2054 (NM_001286342, c.247C>A) was introduced into the OriGene plasmid using a KOD-Plus Mutagenesis kit (Toyobo, Osaka, Japan). The wild-type and mutant *CBY2* sequences were subcloned into the *Nhe*I/*Age*I-digested pDsRed-Monomer-C1 vector (Takara Bio Inc., Kusatsu, Japan). The primer sequences are provided in S1 File. COS7 cells were cultured in Dulbecco's modified Eagle's medium (DMEM) supplemented with 10% FBS. The cells were transfected with the wild-type or mutant vector expressing CBY2 isoform 2 (Q8NA61-2) fused with carboxyl-terminal monomeric DsRed using Lipofectamine 2000 (Thermo Fisher Scientific). At 24 h post-transfection, Hoechst 33342 (R37605; Thermo Fisher Scientific) was added into the culture media according to the manufacturer's instructions, and live-cell imaging was performed using a confocal laser scanning microscope (LSM) 5 Pascal (Carl Zeiss, Oberkochen, Germany).

## In silico functional analysis

Functional annotations of the coding variants were obtained from dbNSFP version 3.0a (https://sites.google.com/site/jpopgen/dbNSFP) using ANNOVAR [15, 22], which provides deleteriousness measures of missense variants, such as SIFT (https://sift.bii.a-star.edu.sg/) and PolyPhen-2 (http://genetics.bwh.harvard.edu/pph2/) [23, 24]. C-scores of the combined annotation dependent depletion (CADD) version 1.2 were obtained from the developer's web server (http://cadd.gs.washington.edu/) [25].

Functional assessment of the splice-site variant was performed using Human Splicing Finder 3 (http://www.umd.be/HSF3/) [26], NetGene2 (http://www.cbs.dtu.dk/services/NetGene2/) [27], NNSPLICE0.9 (http://www.fruitfly.org/seq_tools/splice.html) [28], and SpliceAI (https://github.com/Illumina/SpliceAI) [29].

The 3D structure of the CBY2 protein was predicted using the RaptorX web server (http://raptorx.uchicago.edu/) [30]. The propensity for protein aggregation was predicted using PASTA2.0 (http://old.protein.bio.unipd.it/pasta2/) [31].

## Statistical analysis

The association between the detected candidate variants and IA was assessed by comparing the gene-based variant burden using the two-tailed Fisher's exact test. Candidates for this burden association test were defined as rare and putatively functional variants with allelic frequencies less than 1% in the general population and our controls, and consistently predicted as deleterious by both SIFT and PolyPhen-2 HumDiv [23, 24].

The expression patterns of COS7 cells transfected with CBY2-DsRed were classified into three types according to the study by Seki et al. (2005): (i) cells without aggregation, (ii) with massive aggregations, and (iii) with dot-like aggregations [32]. DsRed-positive cells were counted in three independent fields of view. The difference in the rate of cells with aggregation per field of view between the wild-type and mutant CBY2-DsRed was assessed using an unpaired Student's $t$-test.

A P value less than 0.05 ($p<0.05$) was considered significant in the present study.

## Results

The pedigree F2054 harbored at least seven affected individuals over three generations (Fig 1A), whereas most of the previously studied families were nuclear families for affected sib-pair analysis (Fig 1B and 1C) [10]. Although the genetic basis of F2054 was likely to be complex, involving reduced penetrance, phenocopy, and poly- or oligogenic inheritance, sequence variants shared by all affected participants (III-1, III-4, III-5, and IV-1) were the most highly prioritized in the present study. According to this inheritance scenario, a parametric linkage analysis was performed, assuming III-2 as a carrier and III-6 and 7 as unknown phenotypes. Chromosomal regions spanning over 2cM with positive maximum heterogeneity logarithm of odds (HLOD) scores are depicted in Fig 1D [16, 19]. Variants detected within 582.5cM of these linkage regions were preferentially reviewed in the whole-exome sequencing (WES) data that achieved an on-average 86.0-fold read depth for the exome-enriched regions. Candidate variants were extracted according to each genotype across the seven participants in F2054, which was consistent with the regional IBD-sharing pattern estimated by BEAGLE [20]. This combined method allowed for the systematic elimination of false-positive and false-negative errors occasionally observed in next-generation sequencing (NGS). Table 1 presents seven candidate variants with allelic frequencies less than 1% in the gnomAD database (https://gnomad.broadinstitute.org/), and PolyPhen-2 HumDiv scores greater than 0.957, corresponding to the prediction of probable damage for non-synonymous substitutions (S1 Table in S1 File) [24, 33]. We also confirmed that these variants were not detected in 105 Japanese population-based controls. During the filtering process, variants detected in both III-6 and III-7, which might be considered as intrafamilial controls, were excluded. Other candidate variants validated by Sanger sequencing are listed in S1 Table in S1 File. There were no homozygous or compound heterozygous candidates shared by the affected sisters (III-1, III-4, and III-5) with allelic frequencies less than 3% in the gnomAD and the Human Genome Variation Database (HGVD, https://www.hgvd.genome.med.kyoto-u.ac.jp/) [34]

Among the top seven candidates, the c.1515+1G>A variant in *NPNT* was deleterious because it disrupted the canonical splice donor site. To confirm the *in silico* prediction of aberrant splicing (Fig 2), a minigene assay was performed. The cells expressing c.1515+1G>A demonstrated complete skipping of exon 10, resulting in immediate premature termination p.

**Table 1. Seven representative candidate genes identified from the family F2054.**

| refGene | Sequence change | dbSNP rs ID | Genotype | | | | | | | IBD segment (width cM) Shared members | CADD v1.2 | HGVD ver.1.42 | gnomAD |
|---|---|---|---|---|---|---|---|---|---|---|---|---|---|
| | | | III-1 | III-2 | III-4 | III-5 | III-6 | III-7 | IV-1 | | | | |
| *ALB* [NM_000477] | c.1490T>A (p.V497D) | - | *T/A* | *T/A* | *T/A* | *T/A* | T/T | T/T | *T/A* | chr4:59692718–75681216 (11.13cM) III-1,2,4,5,IV-1 | 14.4 | 0 | 0 |
| *NPNT* [NM_001184692] | c.1515+1G>A (exon10 skipping) | rs776559543 | *G/A* | *G/A* | *G/A* | *G/A* | *G/A* | G/G | *G/A* | chr5:78334484–113356961 (33.07cM) III-1,2,4,5,6,IV-1 | 27.2 | 0 | 0.0002 |
| *ANKRD55* [NM_024669] | c.115G>C (p.D39H) | rs201139565 | *G/C* | *G/C* | *G/C* | *G/C* | *G/C* | G/G | *G/C* | chr5:55352770–60941929 (5.61cM) | 18.77 | 0.0064 | 0.0008 |
| *ACTBL2* [NM_001017992] | c.223G>A (p.G75R) | - | *G/A* | *G/A* | *G/A* | *G/A* | *G/A* | G/G | *G/A* | III-1,2,4,5,6,IV-1 | 14.57 | 0 | 0 |
| *HOXC6* [NM_004503] | c.250C>T (p.L84F) | rs767298048 | *C/T* | *C/T* | *C/T* | *C/T* | *C/T* | C/C | *C/T* | chr12:53153129–55342935 (3.07cM) | 21.7 | 0 | <0.0001 |
| *NCKAP1L* [NM_005337] | c.A248A>G (p.E83G) | - | *A/G* | *A/G* | *A/G* | *A/G* | *A/G* | A/A | *A/G* | III-1,2,4,5,6,IV-1 | 22.3 | 0 | 0 |
| *CBY2* [NM_001286342] | c.247C>A (p.P83T) | rs200515699 | *C/A* | *C/A* | *C/A* | *C/A* | *C/A* | C/C | *C/A* | chr13:40852484–4635951 (7.20cM) III-1,2,4,5,6,IV-1 | 26.6 | 0.0018 | 0.0005 |

IBD segment, identity by descent segments were calculated from the SNP array data using Beagle version 3.3; CADD, C-scores of the combined annotation-dependent depletion version 1.2; HGVD, allelic frequencies in the Human Genome Variation Database release version 1.42; gnomAD, allelic frequencies in the exome dataset of the Genome Aggregation Database. None of the listed variants were observed in our 105 Japanese population-based controls.

Gly450ArgfsTer5 (Fig 3). The gene product Nephronectin (NPNT) is a member of the epidermal growth factor (EGF) repeat superfamily of proteins and has been reported to contribute to endothelial cell activity and angiogenesis [35]. Since impaired endothelial cell function is thought to play a major role in the pathogenesis of aneurysm formation [36], re-sequencing of the entire coding region of *NPNT* was added to 95 familial IA probands. A nonsense variant, c.1364C>A (p.Ser455Ter), was identified in a female patient with aneurysmal SAH. However, a DNA sample from her affected sister was not available because she had died of aneurysmal SAH (Fig 1B).

Of the remaining six missense candidates (Table 1), the *CBY2* variant was of particular interest because the same variant was also shared among affected siblings from another family (F2012, Fig 1C). Moreover, chromosome 13q14.12–21.1 including the *CBY2* locus, has been previously reported as a significant linkage region in a large Caucasian family with IA [37]. In contrast to NPNT, which serves known endothelial cell activities [35], the function of CBY2 in relation to vascular pathophysiology is unknown. Therefore, further analyses were conducted on *CBY2* in the present study.

We first examined the expression of *CBY2* in human cerebrovascular tissues. Immuno-histochemistry demonstrated that CBY2 was expressed in the medial smooth muscle layer of cerebral arterial specimens (Fig 4). RT-PCR further revealed that *CBY2* expression in human brain vascular smooth muscle cells (HBVSMCs) was transcription variant-specific (NM_001286342), which encodes isoform 2 (Q8NA61-2) with a shorter 5' coding sequence compared to isoform 1 (Q8NA61-1) (S1 and S2 Figs in S1 File). Therefore, re-sequencing of the coding exons of isoform 2 was performed in an additional 500 IA probands and 323 non-IA controls, which resulted in the identification of a total of seven nonsynonymous

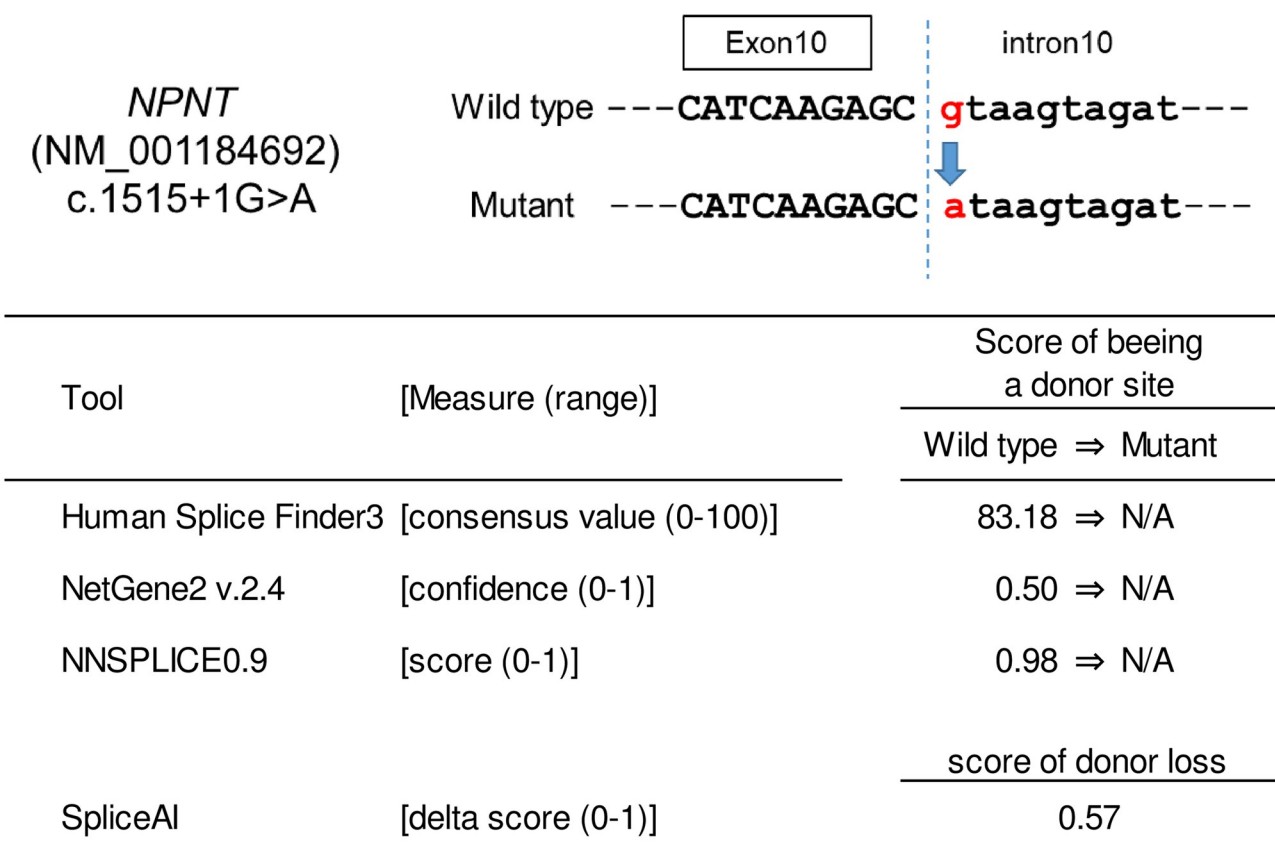

**Fig 2. In silico splicing analysis of the *NPNT* c.1515+1G>A variant.** A functional assessment of the splice-site variant was performed using three web-based and one locally installed programs. All of the programs consistently predict that this variant causes loss of the splice donor site. Abbreviation: N/A, not applicable.

variants. One missense and one truncating variant were common polymorphisms that showed no association with IA ($p = 0.838$ and $p = 0.346$, Fisher's extract test) (S3 Table in S1 File): the c.877A>G (p.Lys293Glu) SNP, predicted as a benign substitution by PolyPhen-2 and SIFT, and the c.749C>A (p.Ser250Ter) SNP, providing a short transcriptional variant lacking the last one of the three consecutive coiled-coil domains. The other five were rare missense variants, of which three were not observed in our 323 controls and were consistently predicted to be damaging by PolyPhen-2 and SIFT (Table 2): c.137G>A (p.Arg46His) was identified in one patient, c.548T>G (p.Leu183Arg) in two patients, and c.247C>A (p. Pro83Thr) in five patients, including the probands of the F2054 and F2012 families. A significant enrichment of these damaging rare variants in *CBY2* among the IA patients (8 of 501) was observed compared to the controls (0/323) ($p = 0.026$, Fisher's extract test).

To explore the functional impact of p.Pro83Thr, wild-type or p.P83T CBY2 fused with carboxyl-terminal monomeric DsRed was transiently expressed in COS7 cells. As previously reported [38], the expressed CBY2 proteins were localized in the cytoplasm (Fig 5). The mutant p.P83T CBY2 exhibited dot-like aggregations more frequently than the wild-type ($p = 0.034$, unpaired Student's *t*-test). This functional change in the aggregation propensity was further supported by *in silico* structural analyses [30, 31]. The p.Pro83Thr substitution elongates the neighboring stretches of the beta strand to form an anti-parallel aggregation motif (S3 Fig in S1 File), reflecting that proline is a well-known secondary structure breaker [39].

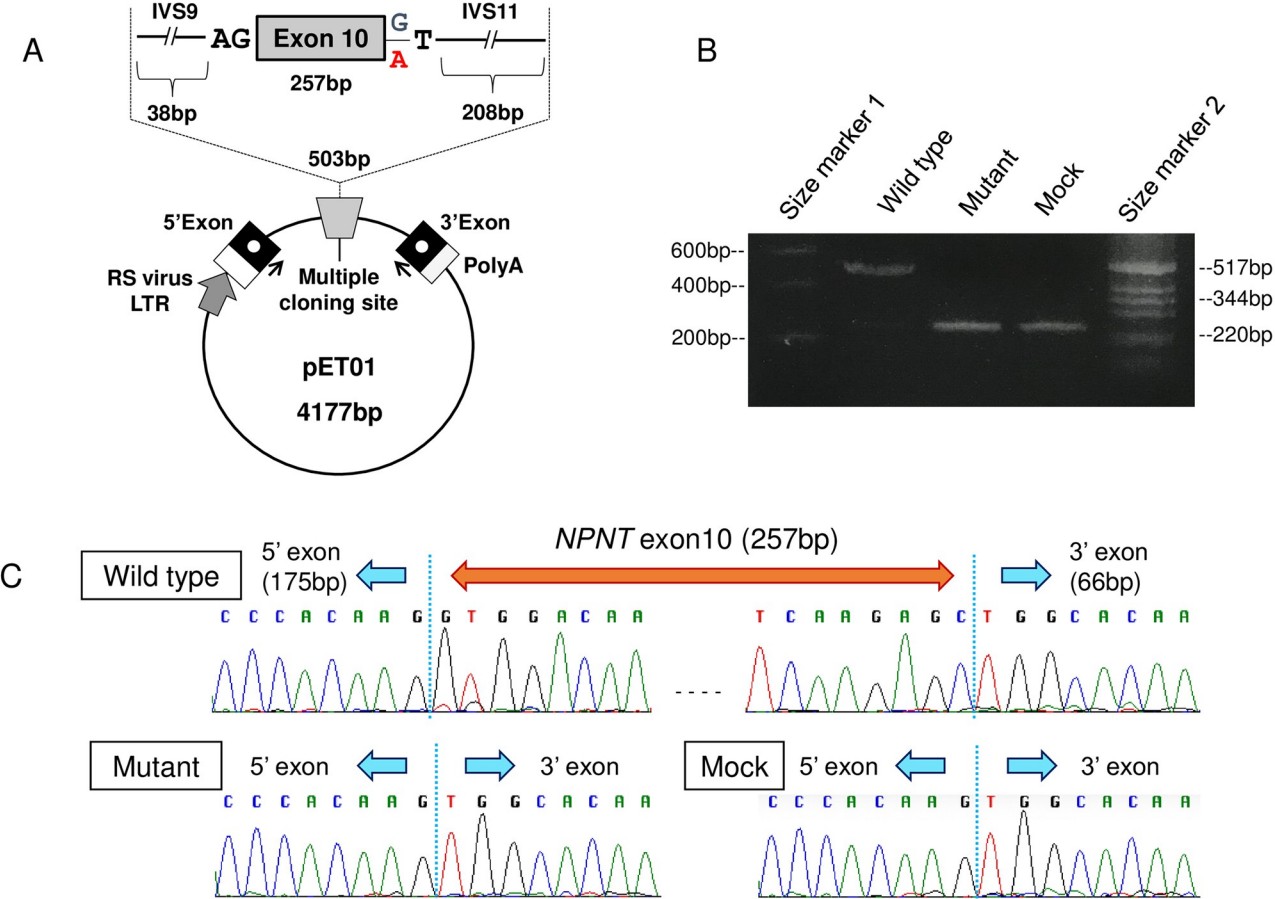

**Fig 3. Minigene splicing assay of the c.1515+1G>A variant in *NPNT*.** (A) The pET01 construct used in this study. The sequences containing the c.1515+1G>A variant in *NPNT* intron 10 (A allele) or those that did not (G allele) were subcloned into the multiple cloning site of the pET01 vector. The arrows under the 3' and 5' exons indicate the primer pair used in RT-PCR after transfection. The primer sequences are provided in S2-1 Table in S1 File. (B) RT-PCR analysis using HeLa cells transfected with the wild-type, mutant, or empty (mock) pET01 vector. The RT-PCR products were separated in a 3% agarose gel electrophoresis and were stained with ethidium bromide. (C) Sequencing chromatograms confirmed the exon 10 was totally skipped in the mutant RT-PCR product.

## Discussion

In the present study, we identified two functional variants in *NPNT* and *CBY2* that segregated with IA in a Japanese multiplex family. The first was the splicing donor c.1515+1G>A variant in *NPNT*, resulting in aberrant splicing, while the other was the c.247C>A (p.Pro83Thr) variant in *CBY2*, which showed an increased aggregation propensity of the gene product in the cytoplasm. These genes were found to be involved in the functions of vascular endothelial and smooth muscle cells, respectively. Their impairment by genetic variants may disrupt the vascular wall integrity and lead to aneurysm formation in this family.

NPNT was originally identified as an extracellular matrix protein that acts as a functional ligand of integrin alpha-8/beta-1 during embryonic development of the kidney [40]. NPNT expression has been reported in a number of embryonic and adult tissues, including blood vessels, indicating its role in embryonic development, as well as maintenance of various adult tissues [41]. Subsequent studies also highlighted its functional aspects as a member of the EGF repeat superfamily proteins and a homolog of epidermal growth factor-like protein 6 (EGFL6) [42]. Many proteins in this family, including EGFL6, are known to promote endothelial cell

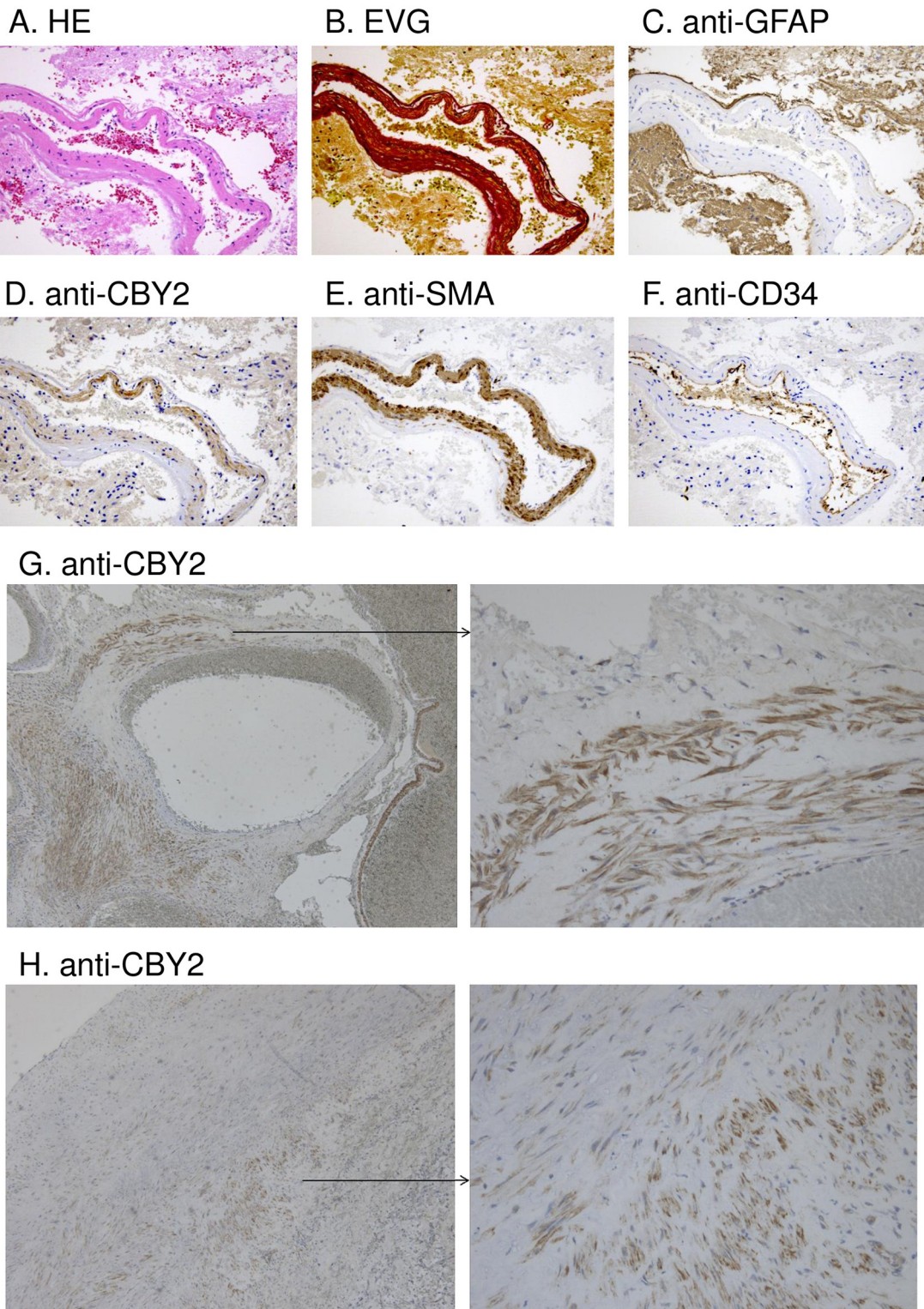

**Fig 4. Immunohistochemistry of cerebral arterial specimens.** The upper six panels show immunohistochemical staining of a surgically resected peripheral cerebral artery from a patient with a brain tumor: (A) hematoxylin-eosin staining, (B) Elastica van Gieson staining (EVG) of elastic fibers, (C) anti-GFAP antibody staining of the glia around the vessel, (D) anti-CBY2 antibody staining, (E) anti-SMA antibody staining of vascular smooth muscle cells, and (F) anti-CD34 antibody staining of vascular endothelial cells. CBY2 is expressed in vascular smooth muscle cells in the tunica media. (G and H) The lower four panels show

anti-CBY2 antibody staining of the surgically resected AVM and IA walls. CBY2 was expressed in smooth muscle cells in the arteriolar wall (G) and was also confirmed in residual smooth muscle cells in the IA wall (H).

migration and angiogenesis [43]. Indeed, a recent study revealed that NPNT had a direct effect on endothelial cell activities and regulated angiogenesis via the extracellular signal-regulated kinase 1/2 (ERK1/2) and p-38 mitogen-activated kinase (MAPK) signaling pathways [35]. In this study, two loss-of-function mutations, c.1515+1G>A, which results in exon 10 skipping, and p.Ser455Ter, were identified in our familial IA patients (Figs 1–3). However, the number of patients with mutations was too small for any conclusions to be drawn, reflecting a high degree of genetic heterogeneity for IA.

In contrast, rare *CBY2* variants were significantly associated with IA in our case-control samples ($p = 0.026$, Fisher's extract test) (Table 2). Three patient-specific deleterious variants (p.Arg46His, p.Pro83Thr, and p.Leu183Arg) were identified in eight IA probands, of which p. Pro83Thr was confirmed to be co-segregated in two pedigrees (Fig 1A and 1C) and caused abnormal protein aggregation in the cytoplasm (Fig 5). Protein aggregation is involved in the etiology of various human diseases due to the loss of protein function, cytotoxicity, or acquisition of novel aggregation-specific functions [44].

The gene product Cby2 (Chibby family member 2) was first identified as an interacting partner of Nek1 (never in mitosis gene A-related kinase 1) in mice [38]. *Nek1*-deficient mice develop facial dysmorphism, male sterility, and slowly progressing polycystic kidney disease (PKD), with renal pathology similar to that of human autosomal dominant polycystic kidney disease (ADPKD), because Nek1 functions in the formation of primary cilia [45, 46]. The family protein Cby1 also plays a crucial role in ciliogenesis by interacting with polycystin-2, a protein mutated in patients with ADPKD [47, 48]. Nek1 phosphorylates tafazzin, an adaptor protein in an E3 ubiquitin ligase complex that targets polycistin-2 for degradation, to maintain normal levels of polycistin-2 for proper ciliogenesis [46]. These lines of evidence indicate that Cby2 is a ciliogenesis-associated protein, although little is known about this molecule apart from its role in intracellular protein trafficking during spermatogenesis in the context of male sterility due to Nek1 deficiency [38].

Primary cilia are found in a variety of cell types, including vascular endothelial and smooth muscle cells, and participate in chemo- and mechanosensing for extracellular signals, such as blood flow, and relaying the signals into the cells [48–50]. A growing body of evidence has demonstrated that primary cilia dysfunction contributes to the development of various

**Table 2. Association analysis with rare sequence variants in *CBY2*.**

| Locus | Nucleotide change | Amino acid change | dbSNP | No. of detected subjects | | | SIFT | PolyPhen2 |
| | | | rs-ID | Familial cases (n = 154) | Sporadic cases (n = 347) | Controls (n = 323) | | HumDiv |
|---|---|---|---|---|---|---|---|---|
| Exon3 | c.137G>A | p.R46H | rs1248037417 | 1 | 0 | 0 | Deleterious | Probably Damaging |
| Exon3 | c.247C>A | p.P83T | rs200515699 | 2 | 3 | 0 | Deleterious | Probably Damaging |
| Exon3 | c.456G>C | p.M152I | rs1370354284 | 0 | 1 | 0 | Tolerated | Benign |
| Exon3 | c.548T>G | p.L183R | rs533443725 | 1 | 1 | 0 | Deleterious | Probably Damaging |
| Exon3 | c.919G>C | p.A307P | - | 0 | 0 | 1 | Tolerated | Possibly Damaging |
| Total no. of deleterious variants | | | | 4 | 4 | 0 | | |
| Burden association test | | | $p = 0.026$ (vs. total cases), $p = 0.011$ (vs. familial cases) | | | | | |

The burden association test was performed using a two-tailed Fisher's exact test.

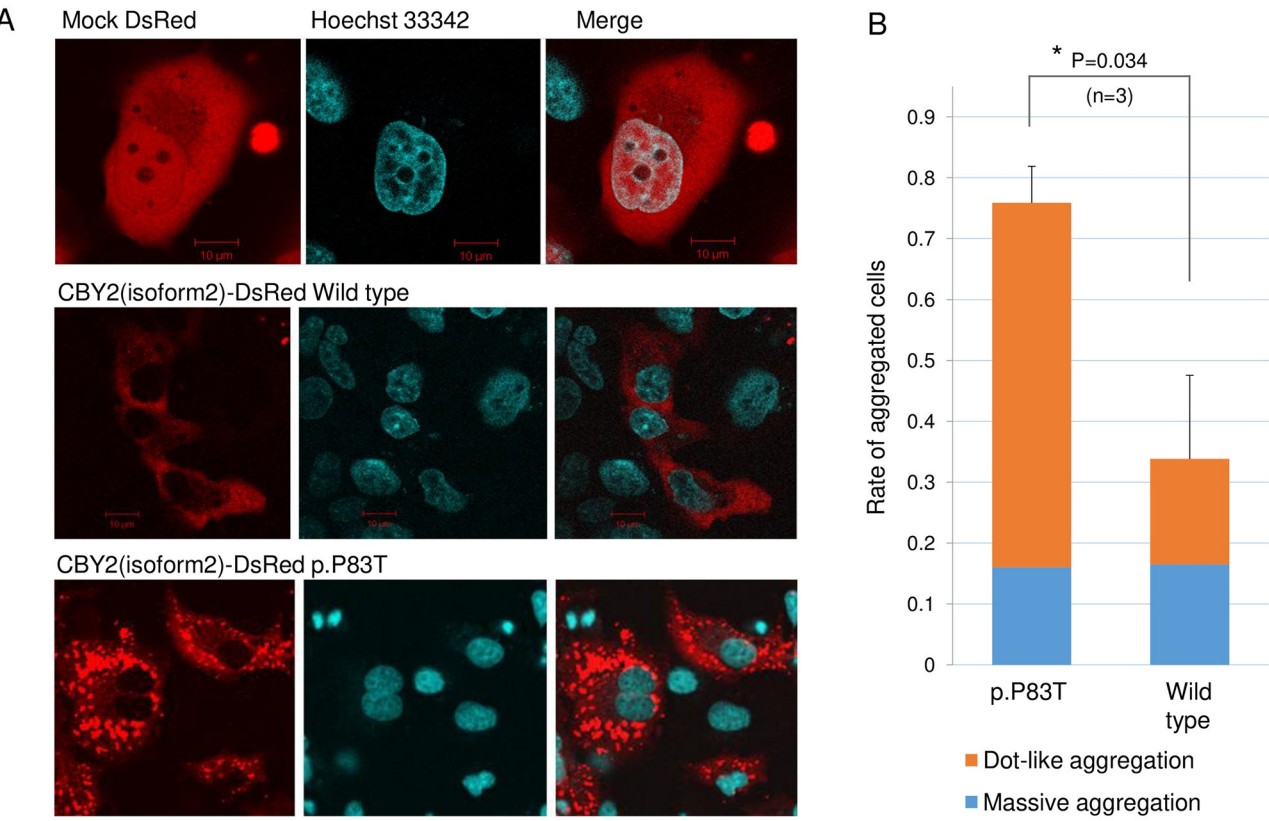

**Fig 5. Cell imaging of CBY2-expressing COS7 cells.** (A) Wild-type or p.P83T CBY2-DsRed was transiently expressed in COS7 cells. The upper three panels represent mock-transfected cells. p.P83T CBY2-DsRed exhibited dot-like aggregations in the cytoplasm. (B) These aggregates were observed more frequently in the cells expressing p.P83T CBY2 than that expressing wild-type CBY2 ($p = 0.034$, unpaired Student's $t$-test). DsRed-positive cells were counted in three independent fields of view.

vascular diseases, such as hypertension, arteriosclerosis, and IA [50, 51]. However, the underlying molecular mechanisms remain to be elucidated. In this study, we confirmed that *CBY2* is expressed in cerebrovascular smooth muscle cells and is significantly associated with IA, which provides insights into the role of the cilia-associated protein network in the pathogenesis of IA.

A potential limitation of this study was that the mode of inheritance prioritized in the exome-sequenced family F2054 was limited. The allele-sharing pattern of the *NPNT* and *CBY2* variants was reasonable as the obligate carrier III-2 was the only member who did not have hypertension, the foremost risk factor for IA, among the seven participants in F2054 (Fig 1A). However, other modes of inheritance should be considered, such as a mode in which susceptibility variants in generations II and III did not transmit to the patients IV-1. For example, a frameshift c.329_330delAC variant (p.Thr111SerfsTer22, rs572295823) in the *CD36* gene (NM_001001548) was detected according this inheritance scenario: it was shared among the patients III-1, 4 and 5, but not transmitted to the patient IV-1 (S1 Table in S1 File). The allelic frequency of this frameshift variant was 0.019% in the East Asian (EAS) population from the 1000 genomes database (https://www.internationalgenome.org/home); therefore, it was regarded as a low-frequency polymorphism and excluded from further analysis in the present study. However, *CD36* is still considered a potential candidate gene. The *CD36* locus (7q21.11) was located within the linkage region of IA (*D7S2415-D7S657*, corresponding to 7q11.22–

21.3), which was previously reported in our affected sib-pair linkage analysis that included patients III-1, 4, and 5 from F2054 [10]. CD36 is a membrane glycoprotein that is expressed in various mammalian cells. On macrophages infiltrating the arterial intima, CD36 acts as a scavenger receptor to internalize oxidized low-density lipoproteins, which induce the secretion of inflammatory cytokines and promote atherosclerosis [52]. As similar macrophage-mediated inflammatory responses have been reported in IA lesions induced in rodent models [53], exploring the association between *CD36* and IA will be a task for future studies. As shown in this example, relaxed thresholds of diseased allele frequencies, penetrance, and phenocopy rates gave a number of candidate variants whose susceptibility to IA could not be determined solely by a single pedigree analysis. Accumulated genetic data from many other large pedigrees with IA will facilitate our understanding of the polygenic architecture of IA susceptibility, which vastly differs from Mendelian inheritance such as autosomal dominant and recessive patterns.

In conclusion, two novel susceptibility genes, *NPNT* and *CBY2*, were identified in this study. The genetic architecture of familial IA seems highly heterogeneous and polygenic, which integrates different etiological mechanisms, such as endothelial dysfunction, primary cilia dysfunction, and atherogenic inflammation. Although an extensive analysis of each multiplex family was only able to detect a few of the leading susceptibility genes within the family, the cross-checking of multiple familial datasets is likely to uncover the diverse and complex genetic causes underlying IA formation.

## Supporting information

**S1 File. This file contains S1-S3 Tables and S1-S3 Figs.**
(DOCX)

**S1 Raw images. Original gel data used in S1 Fig.**
(PDF)

## Acknowledgments

We thank the patients and their families for making this study possible. We also thank Mitsuhiro Amemiya and Akira Saito (StaGen Co. Ltd., Tokyo, Japan) for data processing of next-generation sequencing.

## Author Contributions

**Conceptualization:** Hideaki Onda, Hidetoshi Kasuya.

**Data curation:** Hiroyuki Akagawa, Hideaki Onda.

**Formal analysis:** Tatsuya Maegawa, Hiroyuki Akagawa.

**Funding acquisition:** Tatsuya Maegawa, Hidetoshi Kasuya.

**Investigation:** Tatsuya Maegawa.

**Methodology:** Hiroyuki Akagawa.

**Project administration:** Hiroyuki Akagawa, Hidetoshi Kasuya.

**Resources:** Hideaki Onda, Hidetoshi Kasuya.

**Software:** Hiroyuki Akagawa.

**Supervision:** Hiroyuki Akagawa, Hidetoshi Kasuya.

**Validation:** Hiroyuki Akagawa.

**Visualization:** Tatsuya Maegawa, Hiroyuki Akagawa.

**Writing – original draft:** Tatsuya Maegawa.

**Writing – review & editing:** Hiroyuki Akagawa, Hideaki Onda, Hidetoshi Kasuya.

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
