## [Decision Letter · Decision Letter 0]

28 Dec 2021

PONE-D-21-35037Whole-exome sequencing in a Japanese multiplex family identifies new susceptibility genes for intracranial aneurysmsPLOS ONE

Dear Dr. Akagawa,

Thank you for submitting your manuscript to PLOS ONE. After careful consideration, we feel that it has merit but does not fully meet PLOS ONE’s publication criteria as it currently stands. Therefore, we invite you to submit a revised version of the manuscript that addresses the points raised during the review process.

Please submit your revised manuscript by Feb 11 2022 11:59PM If you will need more time than this to complete your revisions, please reply to this message or contact the journal office at plosone@plos.org. Please include the following items when submitting your revised manuscript:A rebuttal letter that responds to each point raised by the academic editor and reviewer(s). You should upload this letter as a separate file labeled 'Response to Reviewers'.A marked-up copy of your manuscript that highlights changes made to the original version. You should upload this as a separate file labeled 'Revised Manuscript with Track Changes'.An unmarked version of your revised paper without tracked changes. You should upload this as a separate file labeled 'Manuscript'.

We look forward to receiving your revised manuscript.

Kind regards,

Avaniyapuram Kannan Murugan, M.Phil., Ph.D.

Academic Editor

PLOS ONE

“This work was supported by JSPS KAKENHI (grant no. 19K18444 (T.M.), 15K10316, and 18K08953 (H.K.)).”

We note that you have provided information within the Funding Section. Please note that funding information should not appear in other areas of your manuscript. We will only publish funding information present in the Funding Statement section of the online submission form.

 “This work was supported by JSPS KAKENHI (grant no. 19K18444 (T.M.), 15K10316, and 18K08953 (H.K.), https://www.jsps.go.jp/english/index.html).”

“This work was supported by JSPS KAKENHI (grant no. 19K18444 (T.M.), 15K10316, and 18K08953 (H.K.), https://www.jsps.go.jp/english/index.html).”

Additional Editor Comments:

The study is well appreciated. Further, reviewers do support the manuscript yet, raise many concerns and those points to be addressed prior processing the manuscript further. Particularly, the comments from Rev. 1.

Reviewers' comments:

Reviewer's Responses to Questions

**Comments to the Author**

1. Is the manuscript technically sound, and do the data support the conclusions?

Reviewer #1: Yes

Reviewer #2: Yes

2. Has the statistical analysis been performed appropriately and rigorously? 

Reviewer #1: I Don't Know

Reviewer #2: Yes

3. Have the authors made all data underlying the findings in their manuscript fully available?

Reviewer #1: Yes

Reviewer #2: Yes

4. Is the manuscript presented in an intelligible fashion and written in standard English?

Reviewer #1: Yes

Reviewer #2: Yes

5. Review Comments to the Author

Reviewer #1: This study brings forth causal variants in two genes NPNT and CBY2 for intracranial aneurysms. The authors have examined families and singletons and conducted expression studies for these genes. The focus has been on CBY2, within the locus identified in a different population and highly expressed in testis.

I have the following comments about this manuscript:

Major comments:

1. Did the authors consider separate population frequency cut offs for predicted AD AR inheritance for each?

2. What population frequency database were referred to? I would include gnomAD, KAVIAR and any other population specific public database if available.

3. In table 1, include population frequency for the alternate allele from gnomAD/ALFA like in Suppl table S1 etc

4. The authors discuss the possibility of a complex pattern of inheritance including reduced penetrance for the multiplex families. I suggest that the authors include an inheritance pattern (autosomal dominant, recessive etc) explicitly in the Discussion to make their case stronger.

5. Can the authors also include a hypothesis about the functional effect of the variants? In supplementary information, the data for splice site variant is shown for NPNT. It would be good to know if the CBY2 and NPNT variants are acting as loss or function or gain of function mutation in IA.

6. The authors mention 'gene product' on many occasions throughout the manuscript. By that, do they mean mRNA or protein? If the latter, then it needs to be modified in the manuscript.

Minor comments:

1. Correct syntax errors: Ex: Remove g for DNA sample of sister in line 267; Remove gDNA in line 386 etc.

2. I would suggest moving Figure S1 to the main manuscript.

Reviewer #2: I genuinely appreciate the effortful and highly systematic methodologies conducted in revealing novel susceptibility genes for intracranial aneurysms in the Japanese family. I do encourage the authors to conduct this well-elaborated methodology at a higher scale with increasing sample sizes and incorporating other non-Japanese candidates as possible in the near future.

6. PLOS authors have the option to publish the peer review history of their article (what does this mean?). If published, this will include your full peer review and any attached files.

Reviewer #1: No

Reviewer #2: **Yes: **MM

---

## [Author Response · Author response to Decision Letter 0]

23 Feb 2022

RESPONSE TO REVIEWER #1: 

We wish to express our appreciation to the Reviewer for his/her insightful comments, which have helped us to considerably improve our manuscript. 

Reviewer #1: This study brings forth causal variants in two genes NPNT and CBY2 for intracranial aneurysms. The authors have examined families and singletons and conducted expression studies for these genes. The focus has been on CBY2, within the locus identified in a different population and highly expressed in testis. I have the following comments about this manuscript:

Comment 1: Did the authors consider separate population frequency cut offs for predicted AD AR inheritance for each?

Response: Thank you for this comment. We have added the following text (revised text lines 254-257):

“There were no homozygous or compound heterozygous candidates shared by the affected sisters (III-1, III-4, and III-5) with allelic frequencies less than 3% in the gnomAD and the Human Genome Variation Database (HGVD, https://www.hgvd.genome.med.kyoto-u.ac.jp/)”

Comment 2: What population frequency database were referred to? I would include gnomAD, KAVIAR and any other population specific public database if available.

Response: As we responded in comment 1 and presented in the Table S1, the Human Genetic Variation Database (HGVD, https://www.hgvd.genome.med.kyoto-u.ac.jp/) in the Japanese general population was referred to in addition to the gnomAD.

Comment 3: In table 1, include population frequency for the alternate allele from gnomAD/ALFA like in Suppl table S1 etc.

Response: Based on the reviewer’s comment, we have added the HGVD and gnomAD data in Table 1.

Comment 4: The authors discuss the possibility of a complex pattern of inheritance including reduced penetrance for the multiplex families. I suggest that the authors include an inheritance pattern (autosomal dominant, recessive etc) explicitly in the Discussion to make their case stronger.

Response: Based on the reviewer’s comment, we have changed the following text from (revised text lines 408-411):

“The accumulated genetic data from many other large pedigrees with IA will facilitate our understanding of the polygenic architecture of IA susceptibility.”

to

“Accumulated genetic data from many other large pedigrees with IA will facilitate our understanding of the polygenic architecture of IA susceptibility, which vastly differs from Mendelian inheritance such as autosomal dominant and recessive patterns.”

Comment 5: Can the authors also include a hypothesis about the functional effect of the variants? In supplementary information, the data for splice site variant is shown for NPNT. It would be good to know if the CBY2 and NPNT variants are acting as loss or function or gain of function mutation in IA.

Response: Based on the reviewer’s comment, we have added the following text (revised text lines 361-363) and reference to describe the presumed effect of the CBY2 variant :

“Protein aggregation is involved in the etiology of various human diseases due to the loss of protein function, cytotoxicity, or acquisition novel aggregation-specific functions [44]. ”

“44. De Baets G, Van Doorn L, Rousseau F, Schymkowitz J. Increased aggregation is more frequently associated with human disease-associated mutations in neutral polymorphisms. PLoS Comput Biol. 2015;11(9):e1004374.”

The NPNT variants identified in our IA cases were loss-of-function, as mentioned in the discussion (revised text lines 353-355). 

Comment 6: The authors mention 'gene product' on many occasions throughout the manuscript. By that, do they mean mRNA or protein? If the latter, then it needs to be modified in the manuscript.

Response: We strictly adhered to the guidelines for human gene nomenclature established by the HUGO Gene Nomenclature Committee (HGNC) throughout the manuscript, which is also recommended in the submission guidelines of PLOS ONE (https://journals.plos.org/plosone/s/submission-guidelines). The HGNC endorses the use of italics to denote genes, alleles and RNAs to distinguish them from proteins, and recommends that “protein and gene symbols should use the same abbreviation”. They further suggested that proteins should be referenced using non-italicized gene symbols to distinguish them from genes. Please see the following paper from HGNC.

Bruford EA, Braschi B, Denny P, Jones TEM, Seal RL, Tweedie S. Guidelines for human gene nomenclature. Nat Genet. 2020 Aug;52(8):754-758. doi: 10.1038/s41588-020-0669-3. PMID: 32747822; PMCID: PMC7494048.

Comment 7: Correct syntax errors: Ex: Remove g for DNA sample of sister in line 267; Remove gDNA in line 386 etc.

Response: We sincerely apologize for these errors. They have been corrected in the revised manuscript.

Comment 8: I would suggest moving Figure S1 to the main manuscript.

Response: Based on the reviewer’s comment, we moved Figure S1 to the main manuscript as Figure 2.

We wish to thank the Reviewer again for the valuable comments.

 

RESPONSE TO REVIEWER #2:

We wish to express our appreciation to the reviewer for his/her insightful comments, which have helped us to considerably improve our manuscript. 

Reviewer #2: I genuinely appreciate the effortful and highly systematic methodologies conducted in revealing novel susceptibility genes for intracranial aneurysms in the Japanese family. I do encourage the authors to conduct this well-elaborated methodology at a higher scale with increasing sample sizes and incorporating other non-Japanese candidates as possible in the near future.

Response: We thank the reviewer for this important suggestion regarding our future research plan.

---

## [Decision Letter · Decision Letter 1]

1 Mar 2022

Whole-exome sequencing in a Japanese multiplex family identifies new susceptibility genes for intracranial aneurysms

PONE-D-21-35037R1

Dear Dr. Hiroyuki,

We’re pleased to inform you that your manuscript has been judged scientifically suitable for publication and will be formally accepted for publication once it meets all outstanding technical requirements.

Kind regards,

Avaniyapuram Kannan Murugan, M.Phil., Ph.D.

Academic Editor

PLOS ONE

Reviewers' comments:

Reviewer's Responses to Questions

**Comments to the Author**

1. If the authors have adequately addressed your comments raised in a previous round of review and you feel that this manuscript is now acceptable for publication, you may indicate that here to bypass the “Comments to the Author” section, enter your conflict of interest statement in the “Confidential to Editor” section, and submit your "Accept" recommendation.

Reviewer #2: All comments have been addressed

2. Is the manuscript technically sound, and do the data support the conclusions?

Reviewer #2: Yes

3. Has the statistical analysis been performed appropriately and rigorously? 

Reviewer #2: Yes

4. Have the authors made all data underlying the findings in their manuscript fully available?

Reviewer #2: Yes

5. Is the manuscript presented in an intelligible fashion and written in standard English?

Reviewer #2: Yes

6. Review Comments to the Author

Reviewer #2: As discussed before, I would encourage the authors to generalize the results to non-Japanese population via multicenter-based studies.

7. PLOS authors have the option to publish the peer review history of their article (what does this mean?). If published, this will include your full peer review and any attached files.

Reviewer #2: **Yes: **MM

---

## [Editor Report · Acceptance letter]

8 Mar 2022

PONE-D-21-35037R1 

Whole-exome sequencing in a Japanese multiplex family identifies new susceptibility genes for intracranial aneurysms 

Dear Dr. Akagawa:

I'm pleased to inform you that your manuscript has been deemed suitable for publication in PLOS ONE. Congratulations! Your manuscript is now with our production department. 

Kind regards, 

on behalf of

Dr. Avaniyapuram Kannan Murugan 

Academic Editor

PLOS ONE